# Shared medical appointments: Translating research into practice for patients treated with ablation therapy for atrial fibrillation

Monika M. Schmidt[1]*, Joan M. Griffin[2], Pamela McCabe[2], Lynette Stuart-Mullen[2], Megan Branda[3], Thomas J. OByrne[2], Margaret Bowers[4], Kathryn Trotter[4], Christopher McLeod[5]

**1** U.S. Department of Veteran's Affairs, Nashville, TN, United States of America, **2** Mayo Clinic, Rochester, MN, United States of America, **3** Department of Biostatistics and Informatics, Colorado School of Public Health, University of Colorado-Denver Anschutz Medical Campus, Aurora, CO, United States of America, **4** Duke University, Durham, NC, United States of America, **5** Mayo Clinic, Jacksonville, FL, United States of America

* monika.schmidt@va.gov

**Data Availability Statement:** All relevant data are within the manuscript and its Supporting Information files.

## Abstract

### Background

People with atrial fibrillation (AF) have lower reported quality of life and increased risk of heart attack, death, and stroke. Lifestyle modifications can improve arrhythmia-free survival/symptom severity. Shared medical appointments (SMAs) have been effective at targeting lifestyle change in other chronic diseases and may be beneficial for patients with AF.

### Objective

To determine if perceived self-management and satisfaction with provider communication differed between patients who participated in SMAs compared to patients in standard care. Secondary objectives were to examine differences between groups for knowledge about AF, symptom severity, and healthcare utilization.

### Methods

We conducted a retrospective analysis of data collected where patients were assigned to either standard care (n = 62) or a SMA (n = 59). Surveys were administered at pre-procedure, 3, and 6 months.

### Results

Perceived self-management was not significantly different at baseline (p = 0.95) or 6 months (p = 0.21). Patients in SMAs reported more knowledge gain at baseline (p = 0.01), and higher goal setting at 6 months (p = 0.0045). Symptom severity for both groups followed similar trends.

**Funding:** The author(s) received no specific funding for this work.

**Competing interests:** The authors have declared that no competing interests exist.

## Conclusion

Patients with AF who participated in SMAs had similar perceived self-management, patient satisfaction with provider communication, symptom severity, and healthcare utilization with their counterparts, but had a statistically significant improvement in knowledge about their disease.

## Introduction

Atrial fibrillation (AF) is the most common sustained heart rhythm disorder and a major chronic illness epidemic, associated with increased risk of stroke, heart failure, and death [1]. Many of the therapeutic options have a risk for potential life-threatening side effects or complications [1]. Affecting between 2.9 and 6.8 million in the United States for 2015 [2–4], AF places a significant burden on the US health care system [5] and adversely affects patient quality of life [6].

Treatment of AF includes anticoagulation for stroke prophylaxis and typically adjunctive medications or percutaneous cardiac catheter ablation [1]. Catheter ablation is an effective strategy for rhythm control in patients with drug refractory AF and has been shown to improve disease specific quality of life compared to conventional medical therapy [7].

More recently, a number of trials that included structured fitness and weight loss programs have demonstrated a beneficial impact, with decreases in symptom burden associated with AF and improvements in quality of life [8, 9]. These same benefits have been shown for patients who have undergone catheter ablation with aggressive lifestyle modification interventions [8, 9]. Subsequently, recent guideline and consensus documents have stressed the importance of adequate clinical time for teaching effective self-management for AF to assure patients' maintenance of wellness, optimize medical management, and sustain behavior change [10, 11]. While targeted efforts at weight loss, improving fitness, and patient education about the disease process may facilitate the effective management of AF, behavior change is complex and translation of effective interventions into clinical practice is challenging. These interventions are typically time and clinician-intensive and busy practices may not have the resources to adopt guidelines for patient education and consultation about self-management. Identifying whether patient-oriented, cost-effective, sustainable, real-world approaches to lifestyle modification has similar effects to standard approaches to clinical intervention is critical for improving quality outcomes for patients. Shared medical appointments (SMA) are an innovative approach for delivering guideline adherent education, skill development, and disease management into practice. SMAs involve 6–8 patients sharing a 90-minute visit, with an emphasis on education about disease management. Because the appointments are shared, SMAs allow for increased patient education time, coping support from other participants, promotion of lifestyle modification through group sharing and increased patient knowledge of disease process [12]. Rooted in Social Cognitive Theory [13], which emphasizes the influence of the social environment on behavior change, SMAs have been shown to improve chronic disease management [14–17], improve patient satisfaction and self-management, quality of life, and reduce hospitalization [12, 18–23] when compared to standard care. SMAs integrate patient-centered care principles where patients, caregivers and family members actively partner with their clinician or clinical team to develop a care plan that reflects their values and abilities. The goal of SMAs is to activate patients to manage their care effectively by improving their confidence, thus promoting healthier behaviors, improving clinical outcomes and reducing healthcare

utilization [24–26]. Patient activation, defined as a patient's ability to understand and manage the disease process in the complex dynamic of social, cultural, and physical environments has been shown in previous research to influence self-management behaviors for other chronic diseases and may be an important factor in self-management for AF [23, 24].

To date, SMAs have not been studied in the management of AF or among patients undergoing cardiac ablation. Our primary objectives were to determine if perceived self-management and satisfaction with provider communication about AF management differed between patients being treated with ablation who were exposed to a SMA approach and those who received standard care from baseline to 6 months. Secondary objectives were to explore differences between groups for knowledge about atrial fibrillation, symptom severity, and healthcare utilization. As a theoretically-driven quality improvement study, not a randomized clinical trial, the study was informed by the previous trials on improving quality of life using aggressive lifestyle modification approaches, but was specifically designed to test the impact of SMAs on proximal outcomes for complex patients who often require additional time, education and clinical management.

## Methods

### Project design

De-identified secondary data previously collected during a clinical quality improvement project in the Heart Rhythm Services Clinic at a Midwestern academic medical center was used to evaluate the implementation of SMAs into routine clinical practice. Full details of the project design and implementation are reported elsewhere [26]. Survey data were collected from March 2016 through December 2017 and medical record data were extracted by two non-clinical research coordinators with random auditing performed by clinicians who were blinded to the group from which the data came.

For this project, patients were assigned to either standard care or a SMA based on the patient's preferred appointment date. All pre-procedure consultation visits occurred within one week of the scheduled cardiac ablation procedure. Data were collected from patients on their appointment day (baseline) and at 3 and 6 months after baseline. Utilization of services subsequent to patient's ablation procedure was extracted from patient charts to determine potential differences between patients receiving SMAs and standard care.

### Participants

Patients that were 18 years of age or older with AF and had an appointment for evaluation for an AF ablation procedure were eligible for assignment to either SMA or standard care. Patients were excluded from the quality improvement project if they were unable to read and understand English, had documented cognitive impairments, were receiving active cancer treatment, or hemodialysis, had an implanted left ventricular assist device or if, after their appointment, they chose not to have an ablation. Patients receiving active cancer treatment, hemodialysis, or who had an implanted left ventricular assist devices were not included as study participants because: 1.) these patients are not generally suitable for the catheter ablation procedure, and 2.) the additional care required to treat and manage these diseases while also managing care for an ablative procedure would deviate considerably from the standard of care and potentially confound results.

The quality improvement project was reviewed by the Mayo Clinic Institutional Review Board and determined to not represent research. To analyze these secondary data, however, Mayo Clinic Institutional Review Board approval was obtained. Consent was waived because

data was de-identified and patients previously gave consent to have data from their records used for research purposes.

## Intervention

As previously noted, the development of the SMA curriculum, evaluation and implementation processes are detailed in a separate publication [26]. Briefly, patients assigned to SMAs participated in a SMA prior to the scheduled ablation procedure and again at their 3-months post procedure appointment (see Fig 1).

The content of the pre-procedure and 3-month SMA education sessions is described in Table 1. Curriculum development was guided by the European Heart Rhythm Association Consensus document endorsed by the Heart Rhythm Society [6]. A nurse practitioner led the 90-minute session during the initial SMA. For each visit, per billing requirements, patients assigned to the SMA also had a short,10-minute one-on-one visit with a provider where individual questions and concerns were addressed. The SMA was designed to engage the participants and their family members to share their experiences, provide education about AF in an interactive manner, and create an individualized care plan with group support. As shown in Table 1, the curriculum in the initial and 3-month post-procedure SMAs differed and was intended to address the most salient issues for self-management of AF for those particular time points. A curriculum with sample scripts and slide presentation was developed for both the pre-procedure SMA appointment and 3-month post-procedure SMA appointment to assure consistency in the information presented.

## Standard care

As with the SMAs, standard care appointments occurred prior to the scheduled ablation procedure and at 3-months post procedure. Each visit was typically up to 60 minutes long. Patients met with an electrophysiologist, a certified nurse practitioner, fellow, or physician assistant. During the pre-procedure visit, the clinician reviewed results of diagnostic testing, conducted a history and physical examination, delivered patient education about AF, reviewed treatment options, and obtained signed consent for catheter ablation procedure. During the 3-month visit, the electrophysiologist, nurse practitioner, or physician assistant reviewed the results of diagnostic testing, conducted a history and physical examination, delivered patient education, and determined long term anticoagulation and anti-arrhythmic management.

## Outcomes

Table 2 outlines all outcomes and the data collection schedule.

## Instruments and data collection

Consistent with Social Cognitive Theory, the theoretical basis of the intervention, we captured data on the different theoretical domains using different, psychometrically-sound scales. Other scales, such as the AF Effect on Quality of life (AFEQT), capture important aspects of quality of life, but are devoid of domains included in our framework, such as patient activation, knowledge of atrial fibrillation, patient provider communication, and severity of symptoms experienced by patients. These specific domains were important in understanding the ability of the patient and provider to exchange information bidirectionally during the medical encounter in order to evaluate potential differences between standard care and the SMA group.

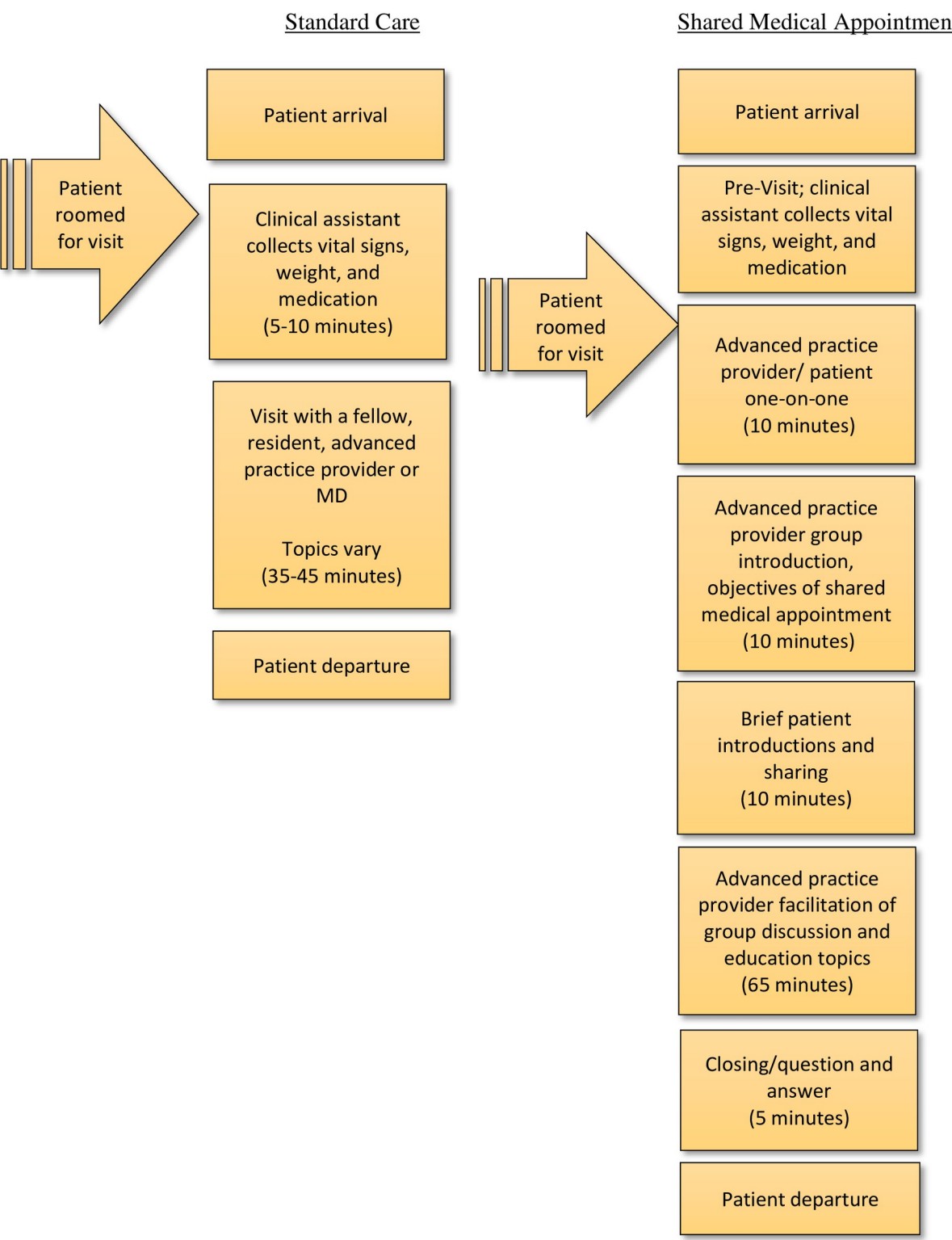

**Fig 1. Pre-procedure and 3-month post-procedure visit flow for patients who underwent cardiac ablation.**

**Table 1. Topics discussed during the shared medical appointments for a cardiac ablation pre-procedure and post-procedure visit.**

| Pre-Procedure SMA | 3-Month Post-Procedure SMA |
|---|---|
| Atrial fibrillation etiology and disease process | Results of post-procedure diagnostic testing |
| Impact of atrial fibrillation on everyday life | Current impact of atrial fibrillation on everyday life |
| Anticoagulation versus bleeding risk: shared decision-making tool | Anticoagulation versus bleeding risk: shared decision-making tool |
| Lifestyle modification research and impact of aggressive lifestyle modification (ie blood pressure, diabetes, cholesterol, weight, alcohol, smoking, and sleep hygiene management) | Follow-up and longitudinal care |
| Treatment options | Creating continuity of care with local providers |
| Technical aspects of catheter ablation | Perceived burden of treatment |
| Risks, benefits, alternatives | Tools for continued lifestyle modification self-management |
| Perceived burden of treatment | Longitudinal symptom management |
| What to expect during hospitalization | |
| What to expect post-hospitalization | |
| Stress management and meditation exercises | |

Perceived self-management of AF was measured using the Patient Activation Measure (PAM)-Short Form at baseline (pre-ablation appointment), 3 and 6 months after ablation [24, 26]. The PAM has demonstrated validity and acceptable reliability when used in populations with chronic illness. The Rasch person reliability for the preliminary 21-item measure was between .85 (real) and .87 (model). Cronbach's alpha was .87 [26]. Scores were categorized into levels of activation with Level 1 indicating low knowledge, motivation, and skill for self-management and Level 4 high motivation, knowledge and skill for self-management. Level 1 scores ranged from 0–47.0; Level 2, 47.1–55.1; Level 3, 55.2–72.4; and Level 4, 72.5–100 [25–27]. PAM data from the baseline, 3-month and 6-month assessments are reported in Table 4.

Patient perceptions of the quality of their healthcare team's chronic disease management, including satisfaction of provider communication about AF management, was measured using the Patient Assessment of Chronic Illness Care (PACIC) [28] and was collected at baseline and at 6 months. The PACIC is a validated tool that assesses indicators of quality that are included in the Chronic Care Model (e.g., activation; delivery system/practice design; goal setting; follow-up/coordination; problem solving) [10]. Psychometric evaluation of the PACIC has shown construct validity and acceptable internal consistency reliability for the entire scale ($\alpha = .93$) and for subscales (ranging from $\alpha = 0.77$ to 0.90)

Patient knowledge about AF was assessed using the Knowledge of Atrial Fibrillation (KAF) test [29]. Because the majority of educational material was presented for the first time in the initial SMA, data were collected prior to and following only the initial SMA or the pre-ablation appointment, depending on the assigned group. Content validity for the KAF has been assessed by expert review and has acceptable internal consistency reliability for the subscales (ranging from $\alpha = 0.64$–0.87) [29].

Self-reported severity of symptomatic episodes were measured using Part C of the University of Toronto Atrial Fibrillation Symptom Severity scale, or AFSS [30]. Response options for AFSS items ranged from not at all to a great deal to assess how often they have been bothered by the symptom in the past 4 weeks. A sum of responses is calculated, with higher scores reflecting greater symptoms severity. Content, construct and criterion validity of the AFSS

**Table 2. Data collection schedule of patient outcomes.**

| Patient Outcomes | Prior to pre-procedure visit | After pre-procedure visit | 3-month post-ablation visit | 6 months post-ablation |
|---|---|---|---|---|
| Confidence of self-management (PAM) | x | | x | x |
| Knowledge of AF & AF treatment (KAF) | x | x | | |
| Satisfaction with provider communication/interaction (PACIC) | x | | | x |
| Symptom Severity (AFSS) | x | | x | x |
| Demographics (marital status, etc.) | x | | | |
| Health care utilization* | | | x | x |
| Dietitian appointment | | | | |
| Physiology appointment | | | | |
| 3-month post-ablation appointment | | | | |
| Triage calls | | | | |
| Hospitalizations | | | | |
| Emergency department visits | | | | |

*Health care utilization between ablation procedure and 3 months and between 3 months and 6 months was extracted from the medical record at 6 months.

have been reported, as has its internal consistency reliability (as 94 α = .72) [30]. This scale has been included in other large-scale studies, including the ARREST-AF study [9], which was intended to allow comparability of findings across studies.

## Electronic medical record data abstraction

As part of the quality improvement project, a medical record review and abstraction was conducted at 6 months post-ablation appointment to assess healthcare utilization, defined as emergency room visits for treatment of AF during the 3-month blanking period and patient-initiated nurse triage calls. Other factors that may affect the relationship between appointment type and outcomes were also abstracted, including chronic comorbidities, body mass index (BMI), smoking status, medication usage and demographic characteristics. Likewise, attendance at appointments with a dietitian and exercise physiologist at the time of the pre-ablation appointment and the 3-month follow-up appointment and the number of hospitalizations and emergency room visits for reasons other than AF were abstracted from the medical record.

## Data analysis

De-identified data from patients that met inclusion criteria, had an ablation, and completed the pre-baseline survey were used for analysis. Baseline characteristics are reported as means and standard deviations for continuous variables, or median and interquartile ranges if data were skewed, and counts and frequencies for categorical data. Outcomes for the data collected at 6 months were modeled as continuous outcomes and adjusted baseline scores using analysis of covariance to account for the correlated nature of the data [31]. Comparisons by assigned condition and utilization were analyzed with the chi-square test. All analyses were conducted in SAS 9.4 (Cary, NC) and p-values reported are based off a two-sided hypothesis. The level of significance was set at p < .05.

## Results

Between March 2016 and December 2017, 216 patients were scheduled for a pre-ablation appointment, and of these, 123 (57%) met inclusion criteria for analysis. Fifty-nine patients

(48%) participated in a SMA and sixty-four (52%) received standard care. Demographic characteristics and other factors at baseline were not significantly different between those in the SMA and standard care groups (Table 3). Patients were, on average, 60 years old, and the majority was male (66%) and non-Hispanic White (99%). Education level and being on an antiarrhythmic medication were different between arms, although not significantly different statistically. As these factors may potentially impact differences in outcomes for standard care and SMAs, analyses for these factors were adjusted in our analyses. Attrition was noted in both groups.

## Patient activation measure

At baseline, both groups' average score was high on patient activation (stage 3), meaning that these patients had reached a point where they were already beginning to make changes and motivated to take accountability for their health [26]. As shown in Table 4, scores between the SMA and standard care groups were not significantly different at baseline (p = 0.73), nor were they different at 6 months (p = 0.21) after adjusting for baseline PAM score, education and antiarrhythmic use.

## Knowledge of atrial fibrillation

Compared to the standard care group, patients in the SMA group reported greater improvement in knowledge of AF scores from pre-to post- appointment after controlling for pre-visit knowledge score, education and anti-arrhythmic use (SMA = 3.6 vs. standard care = 2.2; p = 0.01).

## Patient assessment of chronic illness care

Baseline PACIC scores were not significantly different between the two arms. While the overall PACIC score was not different between groups at 6 months, the goal setting subscale of the instrument showed a statistically significant difference between the two groups (Table 4) after controlling for covariates. On a 4-point scale, patients in the SMA intervention had goal setting scores, on average, 0.7 points higher than those in standard care [95% CI, 0.2, 1.2] (Table 4). None of the other PACIC sub-scales scores were significantly different between assigned conditions at 6 months. For all PACIC subscale scores, the baseline score was found to be a strong univariate predictor for 6-month scores (Patient Activation p = 0.0004, Decision Support p = 0.0001, Problem Solving p = 0.0005, Follow-up p = <0.0001, Goal setting p = 0.0015).

## Atrial fibrillation severity score

At baseline, both groups reported symptoms that placed them in the highest tertile for symptom severity, but scores were not significantly different (SMA = 12.2; standard care = 12.6; p = 0.72). Likewise, at 3 and 6 months post-ablation, both groups experienced an improvement in symptoms, scoring in the moderate tertile (3 months: SMA = 5.1; standard care = 6.1, p = 0.44; 6 months: SMA = 4.2; standard care = 5.9, p = 0.11), yet, again, no significant differences were noted between the groups.

## Utilization

The SMA group had a higher proportion that attended a dietitian appointment than those in standard care (25% vs. 3%, p < .001) (Table 5). No statistical differences were seen in the number who attended exercise physiologist appointments prior to ablation. Of the SMA patients, 78% attended the scheduled 3-month follow-up visit, while only 63% of standard care patients

**Table 3. Demographic and clinical characteristics of the sample N = 123.**

| | SMA (N = 59) | Standard Care (N = 64) | p-value |
|---|---|---|---|
| **Mean age in years** (SD) | 60.7 (10.1) | 59.2 (12.3) | 0.51[1] |
| **Gender** | | | 0.95[2] |
| Female | 20 (33.9%) | 21 (33.3%) | |
| Male | 39 (66.1%) | 42 (66.7%) | |
| **Marital Status** | | | 0.69[2] |
| Married | 47 (83.9%) | 48 (81.4%) | |
| Domestic Partnership | 2 (3.6%) | 2 (3.4%) | |
| Divorced | 4 (7.1%) | 5 (8.5%) | |
| Widowed | 3 (5.4%) | 2 (3.4%) | |
| Never Married | 0 (0.0%) | 2 (3.4%) | |
| **Highest Education** | | | 0.13[2] |
| High School or less | 13 (23.2%) | 7 (11.4%) | |
| Some college | 10 (17.9%) | 23 (37.7%) | |
| Bachelors | 18 (32.1%) | 19 (31.1%) | |
| Graduate School | 15 (26.8%) | 11 (18.0%) | |
| Other | 0 (0.0%) | 1 (1.6%) | |
| **Current Work Status** | | | 0.41[2] |
| Full-time | 29 (49.2%) | 32 (50.0%) | |
| Part-time | 5 (8.5%) | 5 (7.8%) | |
| Retired | 20 (33.9%) | 16 (25.0%) | |
| Not working | 2 (3.4%) | 8 (12.5%) | |
| Other | 3 (5.1%) | 3 (4.7%) | |
| **Annual Household Income** | | | 0.44[2] |
| <$20,000 | 2 (3.6%) | 2 (3.6%) | |
| $20,000 to $29,999 | 6 (10.9%) | 4 (7.1%) | |
| $30,000 to $39,999 | 5 (9.1%) | 2 (3.6%) | |
| $40,000 to $59,999 | 4 (7.3%) | 10 (17.9%) | |
| $60,000 to $79,999 | 6 (10.9%) | 10 (17.9%) | |
| $80,000 to $99,999 | 6 (10.9%) | 7 (12.5%) | |
| $100,000 or more | 26 (47.3%) | 21 (37.5%) | |
| **Insurance** | | | 0.64[2] |
| Government | 16 (27.1%) | 13 (20.3%) | |
| Private | 38 (64.4%) | 44 (68.8%) | |
| Other | 5 (8.5%) | 7 (10.9%) | |
| **Hispanic** | | | 0.31[2] |
| Yes | 1 (1.8%) | 0 (0.0%) | |
| No | 54 (98.2%) | 57 (100.0%) | |
| **Race** | | | 0.33[2] |
| White | 56 (100.0%) | 58 (98.3%) | |
| American Indian or Alaska Native | 0 (0.0%) | 1 (1.7%) | |
| **BMI[3]** | 31.0 (7.0) | 31.0 (7.0) | 0.99[1] |
| **Smoking Status** | | | 0.65[2] |
| Current | 5 (9.6%) | 5 (8.9%) | |
| Past | 14 (26.9%) | 21 (37.5%) | |
| Never | 31 (59.6%) | 29 (51.8%) | |
| Unknown | 2 (3.8%) | 1 (1.8%) | |
| **Mean Systolic BP in mmHg** (SD) | 124.1 (14.5) | 122.0 (15.9) | 0.28[1] |

(*Continued*)

**Table 3.** (Continued)

| | SMA (N = 59) | Standard Care (N = 64) | p-value |
|---|---|---|---|
| **Antiarrhythmic medication** | | | 0.055[2] |
| Yes | 16 (27.1%) | 28 (43.8%) | |
| No | 43 (72.9%) | 36 (56.3%) | |
| **AV Blockade** | | | 0.60[2] |
| Yes | 35 (59.3%) | 35 (54.7%) | |
| No | 24 (40.7%) | 29 (45.3%) | |
| **Anticoagulant medication** | | | 0.14[2] |
| Yes | 49 (83.1%) | 46 (71.9%) | |
| No | 10 (16.9%) | 18 (28.1%) | |
| Diabetes | | | 0.17[4] |
| Yes | 8 (16.3%) | 5 (7.9%) | |
| No | 41 (83.7%) | 58 (92.1%) | |
| Hypertension | | | 0.58[2] |
| Yes | 31 (57.4%) | 32 (50.8%) | |
| No | 23 (42.6%) | 31 (49.2%) | |
| Hyperlipidemia | | | 0.85[2] |
| Yes | 21 (40.4%) | 24 (38.1%) | |
| No | 31 (59.6%) | 39 (61.9%) | |
| Obstructive Sleep Apnea | | | 0.85[2] |
| Yes | 22 (40.7%) | 28 (43.8%) | |
| No | 32 (59.3%) | 36 (56.3%) | |
| Heart Failure | | | 0.28[4] |
| Yes | 1 (2.1%) | 4 (6.3%) | |
| No | 47 (97.9%) | 59 (93.7%) | |

Missing values not shown nor calculated in the percentage.

[1]Kruskal Wallis.

[2]Chi-Square.

[3]One patient missing BMI in SMA Arm.

[4]Fisher's Exact.

attended the follow-up clinical visit (p = 0.06). No difference was found in the average number of triage calls, hospitalizations or emergency room visits from baseline to 6 months between the SMA and standard care groups.

## Discussion

This quality improvement project demonstrates that SMAs for patients with AF did not negatively impact perceived confidence for self-management or satisfaction with patient-provider communication. They did, however, increase knowledge about AF compared to the standard care delivery model. Additionally, we observed that, based on PAM scores, patients in our sample with AF were highly motivated at baseline. Because patient activation in patients with AF has not previously been reported, we are unable to determine how patients in our sample compare to other patients with AF. This level of motivation is clinically relevant given the importance patient activation has on lifestyle and behavior modification for disease process management [11, 24, 25]. We have described how the patient activation in our sample compares to other those of other diseases processes in a separate publication [32].

**Table 4. Patient reports of outcomes collected at prior to pre- catheter ablation appointment, immediately following pre- catheter ablation appointment, 3-months post catheter ablation, and 6-months post catheter ablation.**

| Mean (95% CI) | Baseline | | | 3 –month follow-up† | | | 6-month follow-up† | | |
|---|---|---|---|---|---|---|---|---|---|
| | SMA (N = 59) | Standard care (N = 64) | Mean Difference (95% CI)* | SMA (N = 41) | Standard care (N = 39) | Mean Difference (95% CI)* †‡ | SMA (N = 37) | Standard care (N = 42) | Mean Difference (95% CI)* †‡ |
| **PAM Score** | 67.9 (62.7, 73.1) | 67.7 (63.1, 72.3) | 0.2 (-6.7, 7.1) | 68.1 (59.0, 77.3) | 72.8 (64.3, 81.3) | -4.7 (-13.1, 3.8) | 65.3 (54.3, 76.4) | 71.8 (61.9, 81.8) | -6.5 (-16.6, 3.6) |
| **KAF Pre** | 16.2 (15.2, 17.3) | 16.5 (15.4, 17.6) | -0.2 (-1.7, 1.3) | | | | | | |
| **KAF Post** | 19.8 (18.9, 20.8) | 18.7 (17.8, 19.6) | 1.1 (0.3, 2.0) | | | | | | |
| **AFSS** | 12.2 (10.4, 14.0) | 12.6 (10.8, 14.5) | -0.5 (-3.0, 2.1) | 5.1 (2.3, 7.9) | 6.1 (3.5, 8.7) | -1.0 (-3.6, 1.6) | 4.2 (1.9, 6.5) | 5.9 (3.8, 8.0) | -1.7 (-3.7, 0.4) |
| **PACIC Overall** | 2.7 (2.5, 3.0) | 2.9 (2.7, 3.2) | -0.2 (-0.5, 0.2) | | | | 3.4 (3.0, 3.8) | 3.0 (2.7, 3.4) | 0.3 (-0.04, 0.7) |
| **Patient Activation** | 3.5 (3.2, 3.8) | 3.7 (3.4, 4.0) | -0.2 (-0.7, 0.2) | | | | 4.1 (3.5, 4.6) | 3.9 (3.4, 4.4) | 0.1 (-0.4, 0.7) |
| **Decision Support** | 3.3 (3.0, 3.6) | 3.5 (3.2, 3.7) | -0.2 (-0.6, 0.2) | | | | 3.8 (3.3, 4.2) | 3.5 (3.1, 3.9) | 0.3 (-0.2, 0.7) |
| **Goal Setting** | 2.4 (2.1, 2.6) | 2.7 (2.5, 3.0) | -0.4 (-0.7, 0.01) | | | | 3.7 (3.2, 4.2) | 3.0 (2.5, 3.4) | 0.7 (0.2, 1.2) |
| **Problem Solving** | 3.0 (2.7, 3.4) | 3.2 (2.9, 3.5) | -0.2 (-0.6, 0.3) | | | | 3.7 (3.1, 4.2) | 3.3 (2.8, 3.8) | 0.3 (-0.2, 0.9) |
| **Follow-up** | 2.0 (1.8, 2.3) | 2.1 (1.8, 2.4) | -0.1 (-0.5, 0.3) | | | | 2.2 (1.7, 2.6) | 2.1 (1.7, 2.5) | 0.0 (-0.4, 0.5) |

PAM, Patient Activation Measure; KAF, Knowledge about Atrial Fibrillation; AFSS, Atrial Fibrillation Severity Score; PACIC, Patient Assessment of Chronic Illness Care; SMA, Shared Medical Appointment.

†Adjusted by baseline scores for scale along with patient education level and antiarrhythmic medication use.

‡Difference between Follow-up assessments of SMA–Standard Care.

In light of recent evidence that highlights behavior modification as a predictor of post-ablation success [8, 9], our finding is especially encouraging. We suggest that patients who seek ablation for treatment of AF are appropriate candidates for education in a SMA to learn how to partner with their providers in the prevention and management of symptomatic recurrence.

Patients in SMAs significantly improved patient knowledge about AF compared to standard care without increasing nurse triage calls or emergency room visits. An unintended consequence of SMA attendance was higher rates of attendance at the 3-month post-ablation follow-up visit and to dietician appointments. This suggests that SMAs might engage

**Table 5. Differences in healthcare utilization between patients in SMA and standard care*.**

| | SMA (N = 59) | Standard Care (N = 64) | P-Value |
|---|---|---|---|
| Completed dietitian appointment prior to ablation | 15 (25.4%) | 2 (3.1%) | < .001[1] |
| Completed exercise physiology appointment prior to ablation | 27 (46.6%) | 35 (54.7%) | 0.37[1] |
| Completed 3 Month post-ablation appointment | 46 (78.0%) | 40 (62.5%) | 0.06[1] |
| Average number of triage calls 6 months after pre-ablation appointment | 27 (45.8%) | 32 (50.0%) | 0.64[1] |
| Hospitalizations since pre-ablation appointment | 9 (15.3%) | 10 (15.6%) | 0.95[1] |
| Number of emergency visits since pre-ablation appointment | 9 (15.3%) | 7 (10.9%) | 0.48[1] |

[1]Chi-Square [2]Kruskal Wallis.

*Health care utilization between ablation procedure and 3 months and between 3 months and 6 months was extracted from the medical record at 6 months.

encourage attendance of and engagement in subsequent appointments, which for education and management of complex behaviors may be especially beneficial [25, 26].

In a separate analysis of patient activation of our data published elsewhere we found that patient activation for self-management in our patient population was associated with positive health status and educational attainment [32]. Patients referred to quaternary care centers for consideration of catheter ablation (or who self-refer) often have been exposed to multiple prior encounters with clinicians and are well informed about their care. This may explain the high levels of patient activation noted at baseline. This relatively activated patient group may also explain the lack of variability for PACIC scores, with the exception of the PACIC subscale for goalsetting which is unique to the specific SMA clinical encounter.

Satisfaction with patient-provider communication was not adversely impacted by SMAs. Patients in the SMA intervention demonstrated higher satisfaction with their providers' guidance and support in goal setting, an important component for self-management, especially in a patient group that is activated to achieve clinical outcomes that often rely on their ability to modify lifestyle. This, in combination with recent large randomized clinical trials [8, 9] that point to lifestyle modification as critical in AF management, is imperative when designing a clinical encounter that engages the patient for effective behavior and lifestyle modification.

## Limitations

Because of the unique setting and quality improvement design for our study, our findings may not be generalizable in other settings, either because of the lack of diversity in our study sample, the amount of time spent with patients or because patients who seek evaluation for a procedure at a major academic health center may be more motivated to change than patients at community health centers or those with less pronounced or disruptive symptoms. However, our results do demonstrate the feasibility of implementing an SMA approach in a busy and complex clinical setting.

Our approach underlines the importance of understanding patient readiness for self-management in designing and implementing strategies that target lifestyle modification, however, we are limited by our data sources to test associations with longer term outcomes associated with AF ablation procedure success. This, in part, is due to the nature of the cardiac ablation procedure. The presence of atrial arrhythmias in the early post-ablation period is common because of ablation-related inflammation and healing and atrial arrhythmias during this period are not seen as a true recurrence [33]. In fact, atrial arrhythmias during this early post-ablation period are not associated with worse long-term outcomes or arrhythmia recurrence [33]. Ablation procedure success is best measured up to at least 1 year, but because our data is limited to the first 6 months post-ablation, we are constrained at linking intervention results with longer-term AF ablation procedure success.

## Conclusion

SMAs appear to provide a novel option for comprehensive education in the outpatient clinic for patients with AF. Several outcomes were not significantly different across conditions (e.g., quality of communication with providers and utilization), suggesting that a group approach to education does not compromise patient care and satisfaction. Further research is necessary to assess whether other clinical outcomes, such as BMI, serum lipids, systolic blood pressure, hemoglobin A1c, heart rate variability, metabolic equivalent of task gain, and episodic symptomatic recurrence of AF, can be either equally or better managed through this educational platform with emphasis placed on longitudinal success.

## Implications

With increasing provider burnout [34] and healthcare systems pressured to focus on reducing costs, identifying novel models of care and methods to deliver quality, complex care in efficient and effective ways is imperative. Our project demonstrated that AF patients were highly motivated to manage their own care, and that SMAs may be a method to enhance knowledge of the disease process while providing high quality care that does not impact patient reported outcome measures.

## Supporting information

**S1 Data.**
(XLSX)

## Author Contributions

**Conceptualization:** Monika M. Schmidt, Joan M. Griffin, Pamela McCabe, Lynette Stuart-Mullen, Margaret Bowers, Kathryn Trotter, Christopher McLeod.

**Data curation:** Joan M. Griffin, Megan Branda, Thomas J. OByrne.

**Formal analysis:** Megan Branda, Thomas J. OByrne, Christopher McLeod.

**Funding acquisition:** Joan M. Griffin.

**Investigation:** Christopher McLeod.

**Methodology:** Monika M. Schmidt, Margaret Bowers, Kathryn Trotter, Christopher McLeod.

**Project administration:** Monika M. Schmidt, Joan M. Griffin, Christopher McLeod.

**Supervision:** Lynette Stuart-Mullen, Christopher McLeod.

**Validation:** Megan Branda, Thomas J. OByrne.

**Writing – original draft:** Monika M. Schmidt, Joan M. Griffin, Pamela McCabe.

**Writing – review & editing:** Monika M. Schmidt, Joan M. Griffin, Pamela McCabe, Megan Branda, Thomas J. OByrne, Margaret Bowers, Kathryn Trotter, Christopher McLeod.

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
