## [Decision Letter · Decision Letter 0]

24 Nov 2020

PONE-D-20-10458

Shared Medical Appointments: Translating Research into Practice for Patients treated with ablation therapy for Atrial Fibrillation

PLOS ONE

Dear Dr. Schmidt,

Thank you for submitting your manuscript to PLOS ONE. After careful consideration, we feel that it has merit but does not fully meet PLOS ONE’s publication criteria as it currently stands. Therefore, we invite you to submit a revised version of the manuscript that addresses the points raised during the review process.

Please address comments indicated by the Reviewer.

We look forward to receiving your revised manuscript.

Kind regards,

Elena G. Tolkacheva, PhD

Academic Editor

PLOS ONE

Journal Requirements:

2. Please refer to any sample size calculations performed prior to participant recruitment. If these were not performed please justify the reasons. Please refer to our statistical reporting guidelines for assistance (https://journals.plos.org/plosone/s/submission-guidelines.#loc-statistical-reporting).

3. Please include your tables as part of your main manuscript and remove the individual files. Please note that supplementary tables (should remain/ be uploaded) as separate "supporting information" files.

Reviewers' comments:

Reviewer's Responses to Questions

**Comments to the Author**

1. Is the manuscript technically sound, and do the data support the conclusions?

Reviewer #1: Yes

2. Has the statistical analysis been performed appropriately and rigorously? 

Reviewer #1: Yes

3. Have the authors made all data underlying the findings in their manuscript fully available?

Reviewer #1: Yes

4. Is the manuscript presented in an intelligible fashion and written in standard English?

Reviewer #1: Yes

5. Review Comments to the Author

Reviewer #1: In this retrospective study, Schmidt and colleagues determined if perceived self-management differed between patients who participated in shared medical appointments (SMA) compared to patients provided standard of care. They showed that perceived self-management was not significantly different at baseline or 6 months. Patients in SMAs reported a greater knowledge at baseline and higher goal setting at 6 months with symptom severity following a similar trend. In summary, the findings of the study show that patients who participated in SMAs had similar perceived self-management, patient satisfaction with provider communication, symptom severity, and healthcare utilization with their counterparts but experienced significant improvement in knowledge about their AF.

General Comments

This is an interesting study that determined if perceived self-management differed between patients who participated in SMAs as compared to patients who were provided standard of care. The rationale for the study relates to studies showing that SMAs have been effective at targeting lifestyle change in other chronic diseases and may be beneficial in patients with AF. Overall the manuscript is well-written, and the conclusions are supported by the data presented. However, clarification of the following specific comments will greatly enable the reader to interpret the results and potential implications of the study.

Specific Comments

1. As so many different surveys were used in the study, it is hard to keep track of which ones used to determine differences between the SMA and control group. These should be revised to improve readability and understanding of the study. The benefits of using a comprehensive and AF-specific questionnaire such as AFEQT should be discussed.

2. The rationale for excluding patients with a history of chronic kidney disease on hemodialysis and cancer was unclear.

3. Choosing to evaluate an SMA approach in patients referred for AF ablation therapy may not be optimal. A more appropriate cohort would be a group of patients referred to an AF Clinic in whom the choice of therapy has not already been decided may provide very different results.

4. It is unclear if the primary endpoint of the study used the 3- or 6-month surveys and there appears to be some variability in the number of patients who completed one or both of these.

5. It is a little surprising that there was no difference in patient activation measure or patient assessment of chronic illness. Some explanation for this finding should be provided.

6. It is important to correlate outcomes with success rate of the AF ablation procedure in the two groups.

7. The study is limited to white patients and a tertiary referral center. It’s applicability to patients of different ancestries and non-tertiary referral centers is unclear.

6. PLOS authors have the option to publish the peer review history of their article (what does this mean?). If published, this will include your full peer review and any attached files.

Reviewer #1: No

---

## [Author Response · Author response to Decision Letter 0]

8 Jan 2021

Specific Comments

1. As so many different surveys were used in the study, it is hard to keep track of which ones used to determine differences between the SMA and control group. These should be revised to improve readability and understanding of the study. The benefits of using a comprehensive and AF-specific questionnaire such as AFEQT should be discussed.

Thank you for the opportunity to clarify this important point. We have revised the manuscript to include the following language: 

“Consistent with Social Cognitive Theory, the theoretical basis of the intervention, we captured data on the different theoretical domains using different, psychometrically-sound scales. Other scales, such as the AF Effect on Quality of life (AFEQT), capture important aspects of quality of life, but are devoid of domains included in our framework, such as patient activation, knowledge of atrial fibrillation, patient provider communication, and severity of symptoms experienced by patients. These specific domains were important in understanding the ability of the patient and provider to exchange information bidirectionally during the medical encounter in order to evaluate potential differences between standard care and the SMA group.” Referring to the AFSS scale, “This scale has been included in other large-scale studies, including the ARREST-AF study, which was intended to allow comparability of findings across studies.”

2. The rationale for excluding patients with a history of chronic kidney disease on hemodialysis and cancer was unclear.

We have revised the manuscript to reflect rationale for excluding specific patient groups as follows, “Patients receiving active cancer treatment, hemodialysis, or who had an implanted left ventricular assist devices were not included as study participants because: 1.) these patients are not generally suitable for the catheter ablation procedure, and 2.) the additional care required to treat and manage these diseases while also managing care for an ablative procedure would deviate considerably from the standard of care and potentially confound results.”

3. Choosing to evaluate an SMA approach in patients referred for AF ablation therapy may not be optimal. A more appropriate cohort would be a group of patients referred to an AF Clinic in whom the choice of therapy has not already been decided may provide very different results.

Thank you for bringing this important point to our attention. We have attempted to clarify our decision to include patients seeking ablative therapy as follows: 

“Catheter ablation is an effective strategy for rhythm control in patients with drug refractory AF and has been shown to improve disease specific quality of life compared to conventional medical therapy … More recently, a number of trials that included structured fitness and weight loss programs have demonstrated a beneficial impact, with decreases in symptom burden associated with AF and improvements in quality of life.7,8 These same benefits have been shown for patients who have undergone catheter ablation with aggressive lifestyle modification interventions”

4. It is unclear if the primary endpoint of the study used the 3- or 6-month surveys and there appears to be some variability in the number of patients who completed one or both of these.

Our primary objectives were to determine if perceived self-management and satisfaction with provider communication about AF management differed between patients being treated with ablation who were exposed to a SMA approach and those who received standard care from baseline to 6 months. Secondary objectives were to explore differences between groups for knowledge about atrial fibrillation, symptom severity, and healthcare utilization... Attrition was noted in both groups.” This has been clarified in the manuscript. 

5. It is a little surprising that there was no difference in patient activation measure or patient assessment of chronic illness. Some explanation for this finding should be provided.

We have added the following text for clarity: In a separate analysis of patient activation of our data published elsewhere we found that patient activation for self-management in our patient population was associated with positive health status and educational attainment. Patients referred to quaternary care centers for consideration of catheter ablation (or who self-refer) often have been exposed to multiple prior encounters with clinicians and are well informed about their care. This may explain the high levels of patient activation noted at baseline. This relatively activated patient group may also explain the lack of variability for PACIC scores, with the exception of the PACIC subscale for goalsetting which is unique to the specific SMA clinical encounter.

6. It is important to correlate outcomes with success rate of the AF ablation procedure in the two groups.

We have added the following text for clarity: “…we are limited by our data sources to test associations with longer term outcomes associated with AF ablation procedure success. This, in part, is due to the nature of the cardiac ablation procedure. The presence of atrial arrhythmias in the early post-ablation period is common because of ablation-related inflammation and healing and atrial arrhythmias during this period are not seen as a true recurrence33. In fact, atrial arrhythmias during this early post-ablation period are not associated with worse long-term outcomes or arrhythmia recurrence32. Ablation procedure success is best measured up to at least 1 year, but because our data is limited to the first 6 months post-ablation, we are constrained at linking intervention results with longer-term AF ablation procedure success.”

7. The study is limited to white patients and a tertiary referral center. It’s applicability to patients of different ancestries and non-tertiary referral centers is unclear.

The following has been added for clarity: “Because of the unique setting and quality improvement design for our study, our findings may not be generalizable in other settings, either because of the lack of diversity in our study sample, the amount of time spent with patients or because patients who seek evaluation for a procedure at a major academic health center may be more motivated to change than patients at community health centers or those with less pronounced or disruptive symptoms. However, our results do demonstrate the feasibility of implementing an SMA approach in a busy and complex clinical setting.”

---

## [Decision Letter · Decision Letter 1]

28 Jan 2021

Shared Medical Appointments: Translating Research into Practice for Patients Treated with Ablation Therapy for Atrial Fibrillation

PONE-D-20-10458R1

Dear Dr. Schmidt,

We’re pleased to inform you that your manuscript has been judged scientifically suitable for publication and will be formally accepted for publication once it meets all outstanding technical requirements.

Kind regards,

Elena G. Tolkacheva, PhD

Academic Editor

PLOS ONE

Additional Editor Comments (optional):

Reviewers' comments:

Reviewer's Responses to Questions

**Comments to the Author**

1. If the authors have adequately addressed your comments raised in a previous round of review and you feel that this manuscript is now acceptable for publication, you may indicate that here to bypass the “Comments to the Author” section, enter your conflict of interest statement in the “Confidential to Editor” section, and submit your "Accept" recommendation.

Reviewer #1: All comments have been addressed

2. Is the manuscript technically sound, and do the data support the conclusions?

Reviewer #1: Yes

3. Has the statistical analysis been performed appropriately and rigorously? 

Reviewer #1: Yes

4. Have the authors made all data underlying the findings in their manuscript fully available?

Reviewer #1: Yes

5. Is the manuscript presented in an intelligible fashion and written in standard English?

Reviewer #1: Yes

6. Review Comments to the Author

Reviewer #1: (No Response)

7. PLOS authors have the option to publish the peer review history of their article (what does this mean?). If published, this will include your full peer review and any attached files.

Reviewer #1: **Yes: **Dawood DARBAR

---

## [Editor Report · Acceptance letter]

2 Feb 2021

PONE-D-20-10458R1 

Shared Medical Appointments: Translating Research into Practice for Patients Treated with Ablation Therapy for Atrial Fibrillation 

Dear Dr. Schmidt:

I'm pleased to inform you that your manuscript has been deemed suitable for publication in PLOS ONE. Congratulations! Your manuscript is now with our production department. 

Kind regards, 

on behalf of

Dr. Elena G. Tolkacheva 

Academic Editor

PLOS ONE